# New Approaches to the Circle of Sense and Nonsense

**Bill Seaman** [1,2]

1   Computational Media, Arts and Cultures, The Emergence Lab, Duke University, Durham, NC 27708, USA; bill.seaman@duke.edu

2   Department of Art, Art History and Visual Studies, and Music, Duke University, Durham, NC 27708, USA

**Abstract:** I will briefly discuss the history of research-related projects that Mark Burgin and I worked on together. I will then discuss our joint research related to the circle of sense and nonsense. One paper was entitled *In a search for deeper meanings: navigating the circle of Sense and Nonsense and in turn articulating logical varieties as knowledge illuminators* and the second was entitled *In the Circle of Sense and Nonsense, Including A Mathematic Model of Meaning*. This research represents a bridge between the media arts and sciences (my artwork) as a means of embodying ideas exploring a particular approach to meaning production and related computation, as well as Burgin's concepts related to logical varieties and mathematical models of meaning. I will refer to the full papers and links because they present a very robust and full articulation of the concepts discussed here. In this paper, I will briefly touch on the areas of research, supply short definitions, and refer to the relevant historical publications.

**Keywords:** nonsense logic; sense; poly-sense; logical varieties

## 1. Introduction

This paper explores new approaches to logic, the circle of sense and nonsense, and new approaches to meaning production. It also points out the complementary working relationship between Mark Burgin, a scientist, and Bill Seaman, a media artist and researcher. This bridging between disciplines enables new forms and perspectives of the exploration of a specific branch of knowledge production related to sense, nonsense, and paradoxes. Additionally, I will add to the introduction a pre-history of research-related activities by Burgin and Seaman, where they worked together in different capacities.

I approached Mark Burgin with the idea of publishing a book on Otto Rössler's full career as a Scientist. Rössler is best known for his work on chaos theory. I was introduced to Burgin by Gordana Dodig Crnkovic. Burgin came to work on the Rössler book, adding information-related annotations and later publishing it as part of the Information Series of World Scientific, Volume 15. The title of the book is Chaos, *Information and the future of Physics, The Seaman-Rössler Dialogue with Information Perspectives by Burgin and Seaman* [1]. Rössler was very much ahead of his time with respect to information-related theories and experimental processes and was largely left out of the history of information studies, except perhaps for his work related to chaos. This book included Rössler's articulation of a number of speculative areas, including Cryodynamics and Endophysics, among many other unique theories. Burgin, in annotating the text, articulated how Rössler's research can be seen to fall in relation to a number of important historical theories and to many important books and publications, including his own, perhaps pre-dating some of these publications.

Mark had written a number of very important books in his long history of publishing. I worked on a very in-depth review of Mark Burgin's exquisite work, *Theory of Knowledge: Structures and Processes* [2]. This review was published in *Cybernetics and Human Knowing* and was entitled *A Multi-perspective Approach to A Theory of Knowledge* [3]. I was invited by Burgin to become a member of the program committee for the international conference *Theoretical and Foundational Problems (TFP) in Information Studies*, I formatted this without

bold adding italics where he functioned as conference chair [4]. I was also asked to be one of the Editors of *The Book of Abstracts* for the conference. I curated a 2-day special session entitled *Neosentience, Biomimetics and the Insight Engine 2.0.* Videos of the presented talks were provided later online [5]. Needless to say, I was honored to work with him on this wonderful international conference.

Burgin and I began to have many conversations about each of our interests. He often commented on my creativity—and I on his! These were wonderful, open, highly speculative discussions between me (an artist and media researcher) and Burgin the polymath! Mark read many of my early papers and books and was intrigued by a paper entitled *Nonsense Logic and Re-Embodied Intelligence* [6]. Some interesting ideas were also discussed in my PhD Thesis *Recombinant Poetics—Emergent Meaning as Examined and Explored Within a Specific Generative Virtual Environment, 1999* [7]. Mark and I wrote two papers together. Burgin focused on a mathematical model of meaning in one, and he articulated ideas related to logical varieties serving as knowledge illuminators in the other, pointing to a number of historical papers he wrote and/or collaborated on, some of which were in Russian [8–11]. I highly recommend reading the original papers that articulate his concepts in depth.

I will try to point out some of the salient aspects of this research in this paper, opening up a set of ideas that are in some ways still quite novel.

## 2. Results: In a Search for Deeper Meanings—Navigating the Circle of Sense and Nonsense and, in Turn, Describing Logical Varieties as Knowledge Illuminators

In our previous research papers, we sought to elucidate the terms *sense* and *nonsense as well as* indicate important phenomena in the social environment. We explored the relations and interactions between sense and nonsense with the following three-fold goal: seek elaborations of the formal definitions of these concepts; undertake an investigation of the processes of making sense from nonsense; and establish the mathematical foundations for a sense–nonsense theory. Our goal was to explore the dyad of sense–nonsense through the introduction of the concept of *no-sense*, situated between sense and nonsense, and by extending the concept of *sense* to the concept of *poly*-sense. In addition, Burgin constructed a mathematical model of these concepts in a second paper—*The Circle of Sense and Nonsense, Including A Mathematic Model of Meaning* [12]—and also observed related processes exploring new understandings of logical structures such as logical calculus, logical variety, and logical prevariety.

When we look historically at paradoxes, as well as the understanding of nonsense, we can perhaps begin to acquire a deeper understanding of their meaning as part of human knowledge. Paradoxes and what people have accepted as nonsense historically can bring some profound knowledge to people if they have a mind to acquire a deeper meaning as part of knowledge production. Here, we sought to present a network of ideas that explore new understandings of paradox, sense, and nonsense.

Mark created a series of simple diagrams that might make comprehension easier for non-mathematicians in terms of their potential readings and understandings of our topic areas. We understood that each individual brings their own mindset to the understanding of the world. Our goal was to develop a particular methodology exploring advanced nuanced approaches to both paradox and nonsense by exploring novel tactics through the employment of specific logical tools, including logical varieties and logical prevarieties. Burgin and I were deeply interested in what might be called a pluralistic, multi-perspective approach to knowledge production.

Historically, nonsense has been explored as an opposition to sense, and paradox is opposed to the evidential truth. Our approach sought to illuminate a nuanced, more enhanced form of knowing from both a poetic and artistically oriented set of perspectives by presenting a number of more poetic examples, as well as through mathematics and logic, for which examples were provided by Burgin. This enabled us to extend and re-see the relationality between sense and nonsense with respect to particular contexts.

Here, we brought forward the concept of no-sense, related to both sense and nonsense. Simultaneously, we sought to articulate a perhaps playful new position of poly-sense that was related to the concept of polysemy. This sense of meaning production was tied to my own art practice, which embodied examples of this new understanding. I often sought to incorporate word play and puns in my art practice that could be explored in terms of creating new computational contexts through participant interaction in computer-based media artworks. I provided a punning example of no-sense, which I dubbed *know-sense*. This then helped us define our contemporary sense of poly-logic.

Mark articulated a mathematical model of these new approaches to logic in the following forms: logical calculus, logical variety, logical quasi-variety, and logical prevariety. Mark posited the following:

> To apply logic to dealing with nonsense and studying processes of nonsense transformation into sense, we need sufficiently powerful logical systems. These systems are called logical varieties [10]. They represent the new higher than before level of the development of formal or mathematical logic including many-valued logics (sometimes also called multi-value logics), fuzzy logics, relevant logics, and many other novel logical systems [10].

Burgin provided the following explanation:

> Mathematical theories, such as group theory, category theory or set theory consist of many axiomatic theories, i.e., of many syntactic logical calculi. Systems of these syntactic logical calculi are organized in a definite way, which is formalized by the construction of syntactic logical varieties, quasivarieties, and prevarieties [13].

Mark provided the following definition of logical prevarieties in our paper's appendix, for mathematicians:

Now we can give the exact definition of a syntactic logical prevariety.

Let us take a class **K** of syntactic calculi with a language $L$ of logical expressions and a language $R$ of logical rules of inference and fix a class **F** of partial mappings from $L$ to $L$.

**Definition 10** [9]. A triad A triad **M** = $(A, H, M)$, where $A$ and $M$ are sets of expressions that belong to $L$ and $H$ is a subset of $R$, is called:

(1) a *projective syntactic* (*logical*) (**K,F**)-*prevariety* if there exists a set of calculi $C_i = (A_i, H_i, T_i)$ from **K** and a system of mappings $f_i$: $A_i \to L$ and $g_i$: $T_i \to L$ ($i \in I$) from **F** for which the equalities $A = \cup_{i \in I} f_i(A_i)$, $H = \cup_{i \in I} H_i$ and $M = \cup_{i \in I} g_i(T_i)$ are valid (it is possible that $C_i = C_j$ for some $i \neq j$); each calculus $C_i = (A_i, H_i, T_i)$ is called a *component* of **M** and the set **R(M)** of all calculi $C_i = (A_i, H_i, T_i)$ is called the representation set of **M**.

(2) a *syntactic* **K**-*prevariety* if it is a projective syntactic (**K,F**)-prevariety where all $f_i$ and $g_i$ are inclusions, i.e., $A = \cup_{i \in I} A_i$ and $M = \cup_{i \in I} T_i$;

(3) a *projective syntactic* (**K,F**)-*quasivariety* with the depth $k$ if for any $i_1, i_2, i_3, \ldots, i_k \in I$ either the intersections $\cap_{j=1}^{k} f_{ij}(A_{ij})$ and $\cap_{j=1}^{k} g_{ij}(T_{ij})$ are empty or there exists a calculus $C = (A, H, T)$ from **K** and projections $f$: $A \to \cap_{j=1}^{k} f_{ij}(A_{ij})$ and $g$: $N \to \cap_{j=1}^{k} g_{ij}(M_{ij})$ from **F** where $N \subseteq T$;

(4) a *projective syntactic* (**K,F**)-*variety* with the depth $k$ if it is a projective syntactic (**K,F**)-prevariety and for any $h, k, i_1, \ldots, i_k \in I$ either the intersections $\cap_{j=1}^{k} f_{ij}(A_{ij})$ and $\cap_{j=1}^{k} g_{ij}(T_{ij})$ are empty or there exists a calculus $C = (A, H, T)$ from **K** and projections $f$: $A \to \cap_{j=1}^{k} f_{ij}(A_{ij})$ and $g$: $T \to \cap_{j=1}^{k} g_{ij}(T_{ij})$ from **F**;

(5) a *syntactic* **K**-*variety* with the depth $k$ if it is a projective syntactic (**K,F**)-quasivariety with depth $k$ in which all $f$ and $g$ are bijections, i.e., $A = \cap_{j=1}^{k} f_{ij}(A_{ij})$ and $T = \cap_{j=1}^{k} g_{ij}(T_{ij})$.

The calculi $C_i$ are called the *components* of the variety (prevariety) **M**.

When classes **K** and **F** are not specified such a triad **M** = (*A*, *H*, *M*) is simply called a syntactic logical variety, syntactic logical quasivariety or syntactic logical prevariety depending on what conditions it satisfies [13].

Our goal was to expand the concepts of both sense and nonsense through our concept of poly-sense. Mark had many other writings that explored these concepts through mathematical and logical concepts [8–11], yet our collaboration helped to bridge poetic/artistic approaches with logical and mathematical ones, and our paper was presented more in laymen's terms. We also sought to contextualize these human interactions related to second-order cybernetics and, in particular, the concept of circular causality, which has the human included in an interactive systemic loop.

We presented a brief history of examples of nonsense with the goal of exploring and formalizing a set of potential relations between sense and nonsense. Our text grew quite organically via a dialogue between an information specialist and a mathematician collaborating with a media artist/computational media researcher. This collaboration was in part stimulated by my paper, *Nonsense Logic and Re-embodied Intelligence*. In this paper, I presented the following:

> There is a poignant irony to the fact that the computer, a mechanism entirely predicated on symbolic logic, can be used to explore non-sense as well as illogical and elusive resonant artistic content. A work of art can be seen as an organism- like vehicle of content that is both generated and experienced through interaction [6].

I point out the following related to my artistic work:

> If we look historically at the use of nonsense in literature and other forms of art, we find a fertile realm of creative exploration. How can our understanding of nonsense be applied to the realm of interactive art as well as symbolic logic? Here, Lewis Carroll becomes an interesting subject for investigation in that he both authored texts about logic (as Charles L. Dodgson) as well as texts exploring nonsense. Deleuze states in his book entitled *The Logic of Sense:*

> The work of Lewis Carroll has everything required to please the modern reader: children's books or rather, books for little girls; splendidly bizarre and esoteric worlds; grids; codes and decodings; drawings and photographs; a profound psychoanalytic content; and an exemplary logical and linguistic formalism. Over and above the immediate pleasure, though, there is a play of sense and nonsense, a chaos-cosmose... Deleuze continues: The privileged place assigned to Lewis Carroll is due to his having provided the first great mise en scène of the paradoxes of sense–sometimes collecting, sometimes renewing, sometimes inventing, and sometimes preparing them [14,15].

> One goal of the use of computer systems is to come to better understand ourselves. Computers can function as mechanisms of discourse, enabling the exploration of embodied models made operative through interactive mechanisms. Within this computer-based context, through the exploration of nonsense, one can witness a contrasting critique of sense [16]. The subtle displacement of a particular element from a selected context can actually help to illuminate aspects and/or qualities of functionality [6].

In a number of my artistic works and in my PhD thesis work, I was interested in exploring meta-meaning production, where one could interact in a particular environment and explore the production of meaning through this interaction. This directly connects to second-order cybernetics, where the participant becomes part of an interactive computational circular causality brought about through code authorship, interface design, and the specific loading of a database with media elements and processes. I called this "Re-embodied" intelligence [6,7] and considered it to be a branch of artificial intelligence

based on the writing of Igor Aleksander and Piers Burnett at the time. They provide the following definition in their book, *Thinking Machines, The Search for Artificial Intelligence*:

> Rather than becoming embroiled in the controversies which surround the nature of human intelligence, the practitioners of artificial intelligence have generally chosen to define their goals in empirical or operational terms rather than theoretical ones . . . The researcher simply chooses a task that seems to require intelligence (playing chess say or recognising visual images) and tries to build a machine that can accomplish it [17].

For me, the generative building of a virtual environment in real time, exploring images, music, media behaviors, high-end generative computational processes, and text within a specific generative virtual environment, was clearly an "intelligent" undertaking [7].

Interestingly, Burgin, in talking to Seaman, described their intellectual relationship as being complementary. To achieve our goal, we lay out the beginnings of an approach to the concepts of sense and nonsense and pointed to the fact that the words sense and nonsense have a variety of meanings, being polysemous to begin with. We traced the history of changes related to these two terms' polysemous relationality, starting at first with the notion of a consistent semiotic structure—the sense/nonsense dyad, in which sense and nonsense function as antonyms. Yet, unpacking the meaning potentials and, in part, looking at changes over time in the understanding of nonsense, we came to the necessity of introducing an intermediate concept, *no-sense*, which mediated between sense and nonsense, bringing into being the sense/nonsense triad.

In addition, we discussed context and how meaning arises, learning that sense often emerges in a variety of forms, which is represented by the new concept of *poly-sense*, which, in turn, is a phenomenon similar to polysemy in linguistics. We pointed to the pun related to no-sense, which I dubbed *know-sense* in my punning, playful manner.

Upon analyzing the dynamic features of sense and nonsense, we see that nonsense can house a compression of meanings, some of which both refer to sense and additionally playfully throw off sense. This can generate temporal oscillation between sense and nonsense, housing a cybernetic loop in time enabling one to cycle through these different contextual meaning potentials. The repetition of this cycle corresponds to the *circle of sense and nonsense*, bringing us to poly-sense.

We discussed the discovery of non-Diophantine arithmetic in relation to the mathematical notion of 2 + 2 = 5. We demonstrated that, mathematically, this is a correct expression. More precisely, 2 + 2 = 5 is incorrect (nonsense) in the conventional Diophantine arithmetic and is correct in many non-Diophantine forms of arithmetic [10,18].

We pointed out how specific contexts can be the vehicle of this strange truth. We provided a very simple example of this that has to do with drops of water. If one has one drop of water and adds a second drop, they still end up with a single drop of water, albeit a larger one. Thus, we have 1 + 1 = 1. Here, the example has a paradoxical answer of being both the same one, i.e., one drop of water, and the new one, i.e., one drop of water with a new volume, depending on the frame of linguistic reference or what I call context. Thus, one cycles through these two approaches related to this specific context.

We analyzed a set of dynamic features related to sense and nonsense. We showed that nonsense can house a compression of meanings, some of which seem sense-oriented, while others playfully throw off sense. This punning understanding generates a temporal oscillation between sense and nonsense. We pointed to the nature of second–order cybernetics, where our example houses a cybernetic loop in time enabling one to cycle through these different contextual meaning potentials. Exploring this cycle follows the *circle of sense and nonsense*, bringing us to poly-sense. We provided the following conclusions, in which we sought to

- Obtain results suggesting prospective directions for future research in this area;
- Study psychological processes generating transitions in the sense/nonsense triad and the circle of sense and nonsense;

- Study how the sense/nonsense triad and the circle of sense and nonsense function in science;
- Study how the sense/nonsense triad and the circle of sense and nonsense function in art, music, and literature;
- Study how the sense/nonsense triad and the circle of sense and nonsense function in technology and engineering [13].

My work with Mark Burgin has been exciting and rewarding. He brought deep knowledge from many fields to both our research and our creative discussions.

### 3. Conclusions

Our analysis and perhaps novel formation of the conceptual relationality of sense and nonsense explores the utilization of logic for building a mathematical model of the sense/nonsense triad oriented toward the exploration of dynamical processes in the circle of nonsense and sense. To achieve this, we explored logical varieties and prevarieties as a tool for better discussing differing modalities of sense/nonsense relations [9,10]. In the main two papers discussed, we also provided a compendium of visualizations to articulate the concepts of logical varieties and prevarieties and, in particular, tried to help elucidate these concepts for non-mathematicians.

**Funding:** This research received no external funding.

**Institutional Review Board Statement:** Not applicable.

**Informed Consent Statement:** Not applicable.

**Data Availability Statement:** No new data were created or analyzed in this study. Data sharing is not applicable to this article.

**Conflicts of Interest:** The author declares no conflict of interest.

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
