# Peer review of "New Approaches to the Circle of Sense and Nonsense"

_philosophies, doi:10.3390/philosophies9020040_

Round 1

Reviewer 1 Report

Comments and Suggestions for Authors

Thank you for this paper which clarifies the concepts of sense, non-sense, and no-sense and updating my knowledge. If I understand correctly,

"Sense refers to information or ideas that are rational, logical, and based on evidence or reason and are rooted in objective reality, relying on facts, empirical data, and sound principles. It involves the ability to analyze information, draw logical conclusions, and form well-founded beliefs.

Nonsense refers to ideas or information that lack logical reasoning, coherence, or evidence. It is often characterized by absurdity, confusion, or deception. Nonsense can arise from a variety of sources, including misinformation, deliberate deception, or a lack of critical thinking. In the context of literature, Nonsense is inseparable from sense and exists in relation to the sense.

No-sense often refers to something that is beyond our current understanding or outside the realm of our sensory perception or cognitive comprehension. It’s not that it’s illogical or absurd (which would be nonsense), but rather that it’s currently incomprehensible with our existing frameworks of understanding."

This is very consistent with Burgin's theory of general information that relates knowledge and information to external reality and the mental structures that create a model of the real world in the mental world.

The paper demonstrates the genius of the Late Prof. Mark Burgin and is very timely to reflect upon, given the current chaos in the world where common sense is becoming a rarity.

Author Response

Please see the attachment, this primarily has some copy editing.

In terms of the contents of your suggestions, I think the paper is OK without adding new definitions in each case.

Thanks for looking over the paper.

Sincerely,

Author

Reviewer 2 Report

Comments and Suggestions for Authors

Some minor editing of typos are needed.

Author Response

Here, I believe is a version with the typos fixed.

Thanks for your efforts,

Sincerely,

Author
